# Review of Energy Management System Approaches in Microgrids

**Amrutha Raju Battula** [1] , **Sandeep Vuddanti** [1] **and Surender Reddy Salkuti** [2,*]

1   Department of Electrical Engineering, National Institute of Technology Andhra Pradesh (NIT-AP), Tadepalligudem, Andhra Pradesh 534101, India; amrutharaju.sclr@nitandhra.ac.in (A.R.B.); sandeep@nitandhra.ac.in (S.V.)
2   Department of Railroad and Electrical Engineering, Woosong University, Daejeon 34606, Korea
*   Correspondence: surender@wsu.ac.kr; Tel.: +82-10-9674-1985

**Abstract:** To sustain the complexity of growing demand, the conventional grid (CG) is incorporated with communication technology like advanced metering with sensors, demand response (DR), energy storage systems (ESS), and inclusion of electric vehicles (EV). In order to maintain local area energy balance and reliability, microgrids (MG) are proposed. Microgrids are low or medium voltage distribution systems with a resilient operation, that control the exchange of power between the main grid, locally distributed generators (DGs), and consumers using intelligent energy management techniques. This paper gives a brief introduction to microgrids, their operations, and further, a review of different energy management approaches. In a microgrid control strategy, an energy management system (EMS) is the key component to maintain the balance between energy resources (CG, DG, ESS, and EVs) and loads available while contributing the profit to utility. This article classifies the methodologies used for EMS based on the structure, control, and technique used. The untapped areas which have scope for investigation are also mentioned.

**Keywords:** renewable energy sources; microgrid; energy management system; communication technologies; microgrid standards





## 1. Introduction

Over the last few decades, with an increasing population, the world has gone through an exponential consumption of energy which has led to the depletion of conventional resources like coal, crude oil, and natural gas. The exploitation of these resources has a severe impact on the environment with an increase in greenhouse gases [1,2]. To mitigate these effects, a policy has been adopted by different countries to introduce non-conventional/renewable sources to support the fields of electrification and transportation. In electrification, the existing power grid uses conventional sources for generation and lacks power quality. The poor power quality of supply leads to load shedding and blackouts, thereby interrupting the day-to-day activities of the consumers. The conventional grid uses one-third of the total generation fuel to convert into electricity and, with an eight percent loss in transmission lines of the generated electricity, is used to meet the peak demand that also has a five percent probability of occurring, with reduced reliability [3]. Conventional generation does not utilize the heat produced by itself for any application. These drawbacks of the conventional grid could be compensated with penetration of renewable sources at local areas or distributed generation (DG) there by reducing the transmission losses and maximum utilization of the output including heat generated [4–6]. Integration of dispatchable energy sources like wind and PV introduces the problem of intermittent power generation as they generally depend on climatic and meteorological conditions. A hybrid energy system consisting of storage elements and renewable energy sources is used for the continuous supply of power. The future power grid needs to be intelligent to maintain a reliable supply of economical and sustainable power for consumers [7–10]. To overcome the existing challenges in the grid, a smart grid needs to be adopted which controls the

complex process of power exchange and plans as well for the growing energy demand. The future grid requires the support of communication technologies and local microgrids (MG) for efficient control of the system. The integration of renewable energy resources at the load side requires a two-way flow of power and data with the capability of adapting to management applications that can leverage the technology [11]. During a fault condition, the local microgrid isolates itself from the main grid, creating a standalone/islanding mode of supply to the consumers [12,13]. This feature is known as plug and play, which allows the local generation to meet the demand by balancing the energy available. The microgrid consists of a microgrid control center (MGCC) and local controllers (LCs) to balance the energy demand. The microgrid takes the inputs from forecasted parameters (weather, generation, and market prices) to meet the uncertain load demand and also participates in the energy market. The MGCC is supported by communication technologies and equipped with processing algorithms to overcome the challenges in the generation–demand balance [14–17]. The energy management in microgrids controls the power supply of storage elements, demand response, and local controllers/local generation sources. Figure 1 shows a typical structure of a microgrid.

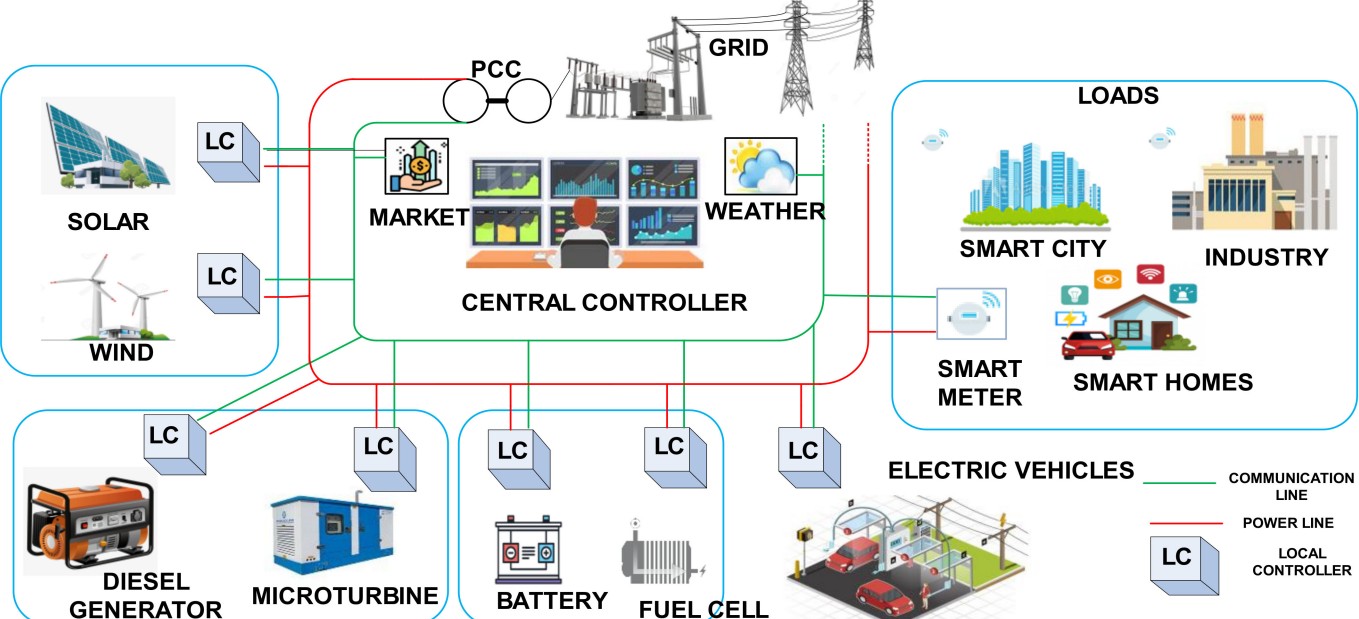

**Figure 1.** Structure of a typical microgrid.

The contributions of this paper are shown as below:

- This paper provides a brief introduction about the architecture of microgrids, different classifications in microgrids, components of a microgrid, communication technologies used, standards available for the implementation, and auxiliary services required.
- This paper provides a review of the recent analysis of the different energy management strategies consisting of classical, heuristic, and intelligent algorithms. The article analyzes each approach and its applications in that methodology.
- The paper addressed applications in energy management which include forecasting, demand response, data handling, and the control structure.
- This article provides insight on areas in which the scope of research and their contribution to energy management is in the nascent stage.

The energy management strategies proposed for the microgrid in the paper are structured into six sections. Section 1 is the introduction to microgrids and energy management. Section 2 provides a brief overview of microgrid elements, architecture, classification, and communication. Section 3 gives an overview of different control structures in energy

management. Section 4 provides reviews on different numerical algorithms used in energy management strategies in microgrids based on the classification, control, and methods of approach. Remarks on each paper for different controls of the EMS application are given. Section 5 discusses the support infrastructure of microgrids for their efficient operation. Section 6 provides the conclusion of the paper.

## 2. Overview of Microgrid

### 2.1. Microgrid Components

A microgrid is a small or medium distribution system comprised of smart infrastructure capable of maintaining equilibrium in demand–supply while providing security, autonomy, reliability, and resilience. Sourced distributed generations (DGs) like photovoltaics (PV), wind turbines (WT), microturbine (MT), fuel cells (FC), and energy storage units (ESU) are expected to deliver electricity without interference from the main grid. This high penetration of DGs can cause challenges in the performance of power system stability in large areas. To minimize the risks, the concept of microgrids is proposed [18,19]. A microgrid is a small-scale low- or medium-level voltage distribution system consisting of distributed energy resources (DERs), intermittent storage, communication, protection, and control units that operate in coordination with each other to supply reliable electricity to end-users [20].

### 2.1.1. Distribution Generations (DGs)

Conventional generation (CG), such as coal-based thermal power plants, hydro power plants, wind-generation farms, and large-scale solar and nuclear power plants, are centralized to supply electricity for long distances. A decentralized generation is energy generated by the end-users by using small-scale energy resources [21,22]. Local generation when compared with the conventional power system reduces the transmission losses and the cost associated with it. The generation could be from 1 kW to a few 100 MW; the generation units are mostly used to support the peak load of the demand. Distributed generation sources consist of both renewable and non-renewable sources, i.e., wind generators, PV panels, small hydro power plants, and diesel generators [23]. Combined heat and power (CHP) is where heating is added along with electricity in the application. The sources that are being used in CHP systems are Stirling engines, internal combustion engines, and micro-turbines (MT) using biogas, hydrogen, and natural gas [24]. CHP technology stores excess allowing optimum performance, thereby attaining efficiency of more than 80%, to that of about 35% for centralized power plants [25]. Table 1 shows characteristics of distributed generation sources.

**Table 1.** Characteristics of distributed generation sources.

| Characteristics | Solar | Wind | Micro-Hydro | Diesel | CHP |
|---|---|---|---|---|---|
| Availability | Location-Based | Location-Based | Location-Based | Anywhere | Source-Based |
| Output | DC | AC | AC | AC | AC |
| Carbon emission | Nil | Nil | Nil | High | Source-Based |
| Interface | Converter | Converter + IG/SG | IG/SG | Generator | Generator |
| Flow control | MPPT/DC Voltage | MPPT/Torque and Pitch | Controllable | Controllable | AVR and Governor |

DC—Direct current, AC—Alternate current, MPPT—Maximum power point tracking, AVR—Automatic voltage regulator.

### 2.1.2. Energy Storage System (ESS)

Energy storage is a device that is capable of converting the electrical energy to a storable form and converting it back to electricity when it is needed. Based on the form of stored energy, there are four main categories for energy storage technologies: mechanical energy storage (MES), thermal energy storage (TES), chemical energy storage (CES), and electrical energy storage (EES). The key components for the working of MG EMS are the energy storage units, which regulate the supply–demand balance during the operation of DGs. In [26–28], a conclusion is drawn that a system with several micro sources is modeled

to support an island mode where storage systems are needed to maintain the balance of the intermittent sources. The energy storage devices that are included in microgrid systems that provide continuous power supply are batteries, flywheels, and supercapacitors [29]. In terms of the current economy, batteries are less expensive and have a high negative environmental effect compared to other storage devices. Storage in fuel cells is also another option that converts the fuel into electricity through a chemical process. These fuel cells require oxygen and hydrogen for continuous supply without discharge. A variety of fuels available for the fuel cell are propane, natural gas, anaerobic digester gas, methanol, and diesel hydrogen [30], while hydrogen has become prominent in recent years for its clean and safe operation. Table 2 shows commonly used energy storage and their characteristics.

**Table 2.** Different energy storage systems in microgrids.

| Characteristics | Charge/Discharge Rate (MW) | Discharge Duration | Response Time | Energy Density (Wh/kg) | Power Density (W/kg) | Environmental Impact | Service (Years) | Efficiency (%) |
|---|---|---|---|---|---|---|---|---|
| Battery | 0–40 | msec–hours | msec | 10–250 | 70–300 | High | 5 | 70–90 |
| Flywheel | 0.001–0.005 | msec–1 h | msec | 0.005–5 | 500–10,000 | Low | 20 | 75–95 |
| Supercapacitor | 0.002–0.25 | msec–15 min | instantaneous | 5–130 | 400–1500 | Low | >10 | 90–95 |
| Fuel Cell | 0.001–50 | sec-day+ | m sec | 800–10,000 | 500–1000 | Moderate | >15 | 20–90 |
| CES | 0.1–300 | Hour–day+ | min | 3–60 | - | Low | 15 | 40–90 |
| SMES | 0.1–10 | msec–10 sec | instantaneous | 0.5–5 | 500–2000 | Low | 10 | >95 |
| Pumped storage | 0.1–5000 | Hour–day+ | Sec–min | 0.5–1.5 | - | Low | 25 | >85 |

### 2.1.3. Loads and Their Classification

Loads can be categorized as residential, commercial, industrial, and others (agriculture and public offices) from the statistical data of feeder consumption in the distribution system. Measurement-based and component-based approaches are considered for load model identification [31]. The measurement-based approach needs the measured data from the smart meters or measuring devices which derives into load model structure. The capturing of data for load characteristics needs to be composed of different environmental conditions. The data obtained from the smart devices are used to form the load model structure as static, ZIP (constant impedance-resistive components or heating, constant current-street lighting, and constant power motors), and exponential [32,33]. Then, the structure is estimated and validated with field measurements by correcting the errors using intelligent detection techniques (artificial intelligence and pattern detection). The component-based approach aggregates the load model by combining the load consumption of individual components, acquired by the information or rating of each load in the load composition. This approach needs three different datasets: (i) individual component load model, (ii) percentage of each component's load consumption, and (iii) share of the load contribution from each load class—residential, commercial, and industrial. The individual component model parameters are obtained from experiments [34–37]. Figure 2 has shown different loads classification is based on identification and control.

The above-discussed techniques and classifications are the key structures for the smart loads. Smart loads are energy-efficient sensor-based controllable load infrastructures that have real-time access to energy usage data. Smart houses control the appliances according to users' preferences, using the intelligence of the appliances to enable the consumer to use real-time energy budgeting to manage in any given day, which allows smart loads to tune the consumer's energy consumption to their daily lifestyle consumption [38,39].

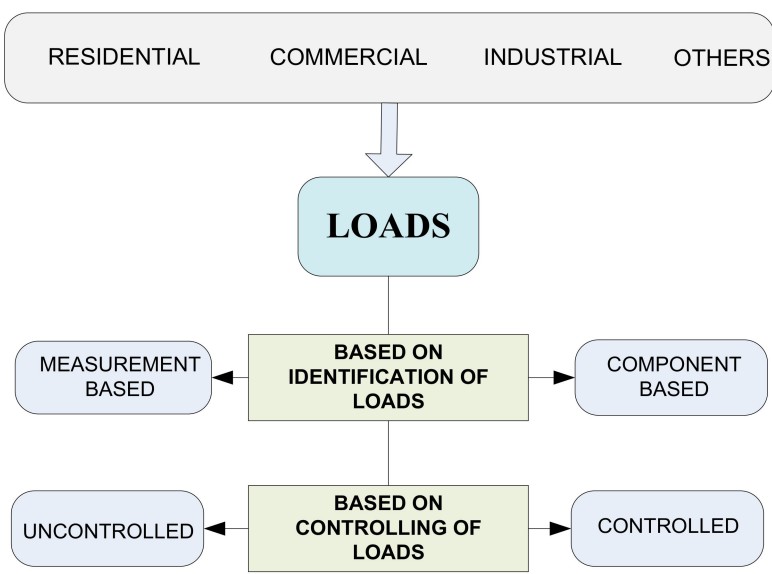

**Figure 2.** Loads classification is based on identification and control.

### 2.1.4. Integration of Electric Vehicles

Increased pollution led the world to move away from conventional fossil-powered vehicles to electric vehicles. Electric vehicles have untapped potential in both environmental and energy applications. A few of the applications of the electric vehicle are the vehicle-to-grid (V2G), vehicle-to-vehicle (V2V) supply of power [40,41]. The connection of EV connected to the grid through the charging station is shown in Figure 3.

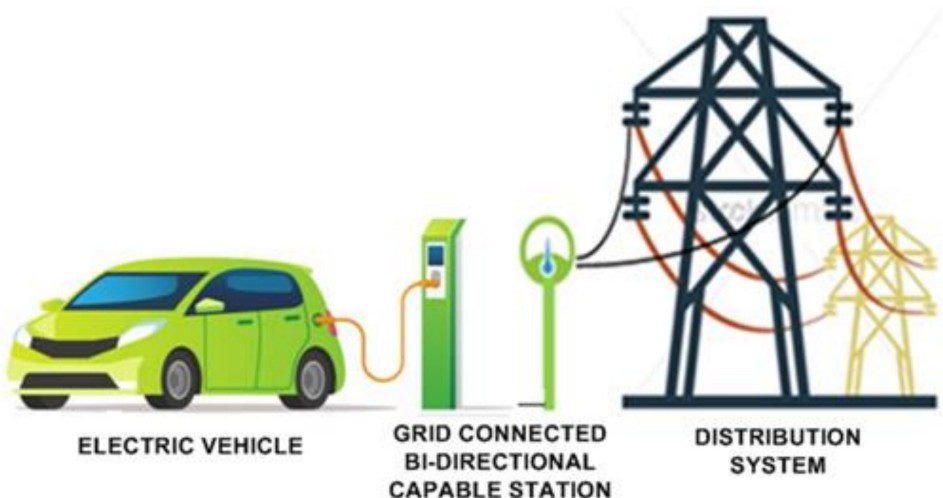

**Figure 3.** Electric vehicle connected to a charging station.

V2G is a process where an electric-powered vehicle supplies power to the regional local grid to meet the demand during peak demand or participate in the energy market by reducing the overall cost of bidding during peak rise in the price of power. This requires communication with the power grid to return the electricity or by controlling the charging rate which enables the EV to support the renewable energy sources from fluctuating, as they cannot be governed [42]. A few of the EVs that support the V2G are battery electric vehicles (BEV), plug-in hybrid vehicles (PHEV), and fuel cell electric vehicles (FCEV). When the electric car batteries are not in use, they can be used to provide electricity to the grid or to charge other storage devices. With an estimated increase in usage of electric vehicles in the future, it is assured to improve the storage capability to balance the demand–supply of the MG. Thus, it provides improved performance in the stability and reliability of the system.

## 2.2. Classifications of Microgrids

A microgrid is generally connected to the grid at the point of common coupling (PCC) through STS (static transfer switch), where voltage and frequency stability is managed by the power grid. When disturbance or failure in the grid occurs, MG maintains the system stability by isolating itself from the main power grid, forming an islanded condition. The renewable energy sources (solar, hydro, wind, and bio), which are not continuous, are connected through power electronic converters (PEC) for good power quality of output; these converters provide a resilient, reliable, continuous, and efficient power supply [43,44]. By the nature of the output obtained, MGs are classified into AC source microgrid, DC source microgrid, and (AC/DC) hybrid microgrid.

An AC microgrid is a common topology of its flexible voltage level transmission using transformers. An AC supply bus is introduced where all DERs, either with DC or AC sources, are connected using PECs to AC loads [45,46]. Almost all the loads in the power system are of AC nature; AC-MG is most sorted. Figure 4 represents a structure of an AC microgrid.

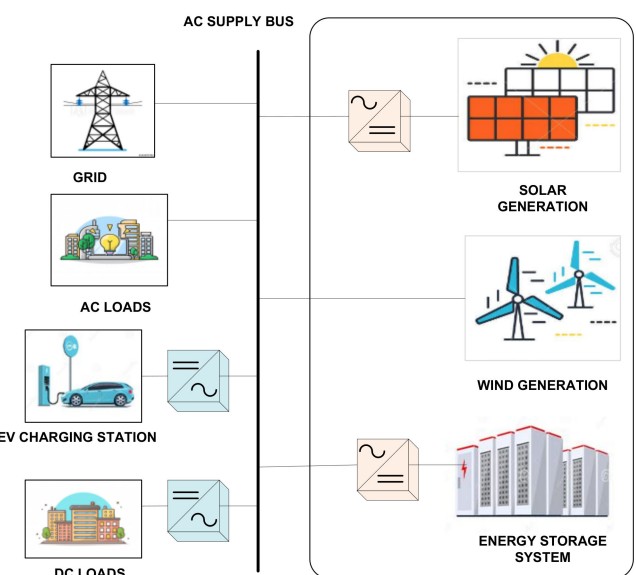

**Figure 4.** Structure of an AC microgrid.

In the DC-MG network, a DC bus connects both AC and DC sources from where the output is taken by the loads [47]. The concept to supply the DC supply is to reduce the number of PEC used, as the DC sources are more available compared to AC sources, which also eliminates the possibility of harmonics due to PEC, as it is not present in DC supply [48]. Increasing popularity in the usage of DC sources like mobiles, laptops, and also household items for isolated places instigated the DC-MG into existence. Figure 5 represents a typical DC microgrid structure.

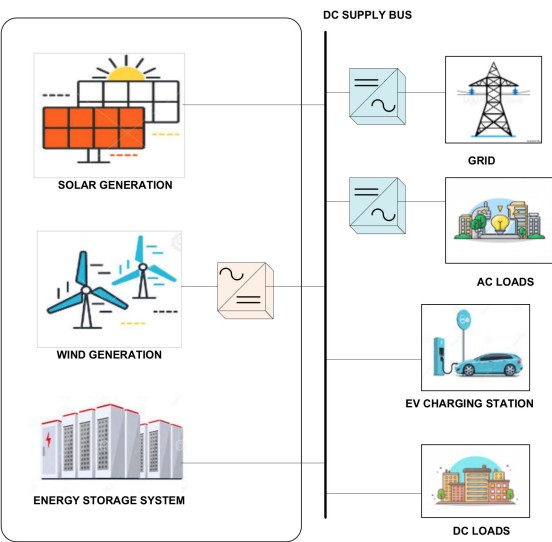

**Figure 5.** Structure of a DC microgrid.

An AC/DC hybrid MG is proposed to effectively introduce both AC and DC sources and consumers in a system. AC sources and DC sources are connected to their respective buses where the outputs are given to the consumers accordingly [49]. The idea of AC/DC hybrid MG is to simultaneously use the supply from both DC and AC sources and thereby reduce the overall power consumption [50]. This is possible by the PEC at both supply buses that support the bi-directional exchange of power from source end to load and vis-à-vis. Figure 6 represents a hybrid microgrid.

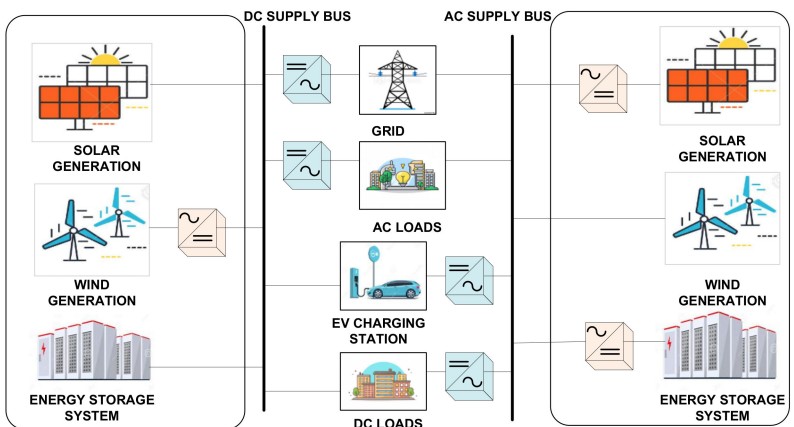

**Figure 6.** Structure of a hybrid microgrid.

*2.3. Control Structure of a Microgrid*

As a small-scale electrical distribution network, an MG has many variables and constraints to control. An energy management system plans, supervises, and manages the system's supply–demand balance while assuring dependable, cost-effective, and efficient operation [51–53]. The management of a microgrid needs to deal with different technical and economical areas, timescales, and infrastructure levels, which requires a control structure to operate the variables. One such control structure for the microgrid is the hierarchical control scheme, which is a generally accepted standardized solution [54].

The hierarchical control structure consists of three different levels operating with individual operating time, data inputs, and control equipment. The different levels in hierarchical control schemes are: (i) primary level, which supervises the control of the DER units; (ii) secondary level, which is responsible for the voltage and frequency modification of the system in coordination with the primary level; (iii) tertiary level, which is the core

control of the system like demand–supply management, storage management, renewable integration, power flow control, optimization of parameters, and control strategies. The tertiary level can also be termed as the energy management system [55]. Figure 7 shows a typical hierarchal control of a MG.

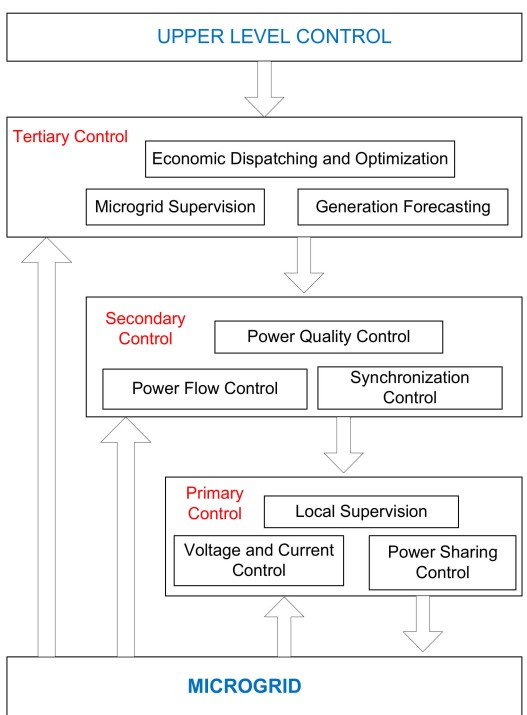

**Figure 7.** Hierarchal control of a microgrid.

### 2.4. Communication of the Microgrid

Communication is an important tool that converts the conventional power network into an intelligent system, connecting generation, transmission, distribution, and utilization systems to the central management center to maintain stability by processing the real-time data. There are several wires and wireless technologies available in the market but the selection of technologies depends on features like data rate, latency, coverage area, reliability, and consumption of power [56]. Table 3 presents various communication technologies used in microgrid. Communication equipment could increase the MG implementation cost with an increased number of communication devices like the repeater and routers for feasible and fast co-collection of data in an area. Increase data collection by the sensors and monitors in the smart homes and smart cities to compensate for the cost and the dire need to reduce the communication infrastructure while maintaining the reliable operation [57,58]. With recent trends in MG's integration and to incorporate internet of things (IoT) devices for measuring, it is better to consider wireless communication technology for its wider applications [59].

**Table 3.** Communication technologies in a microgrid.

| Technology | Spectrum | Data Rate | Range |
|---|---|---|---|
| GSM | 900–1800 MHz | 14.4 Kb/s | 1–10 km |
| GPRS | 900–1800 MHz | 170 Kb/s | 1–10 km |
| 3G | 1.92–1.98 GHz | 2 Mb/s | 1–20 km |
| 4G | 2.11–2.6 GHz | 100 Mb/s | 1–10 km |
| 5G | 3–90 GHz | 10 Gb/s | >1 km |
| WiMAX | 2.5–5.8 GHz | 75 Mb/s | 10–50 km |
| PLC | 1–30 MHz | 2–3 Mb/s | 1–3 km |
| Zigbee | 800 MHz–2.4 GHz | 250 Kb/s | 30–50 m |
| Bluetooth | 2.4–2.483-GHz | 2.1 Mb/s | 0.1–1 km |

## 3. Energy Management System Control Structure

### 3.1. Structure of EMS

According to the International Electro-Technical Commission (IEC) standard application program about power systems, IEC-61,970 defines an energy management system as a "computer system comprising a software platform providing basic support services and a set of applications providing the functionality needed for the effective operation of electrical generation and transmission facilities to assure adequate security of energy supply at minimum cost" [60].

Different operations of EMS are data analytics, forecasting, optimization, and human–machine interface (HMI), and network reconfiguration for real-time interface with the EMS. Figure 8 shows the structure of the EMS of an MG.

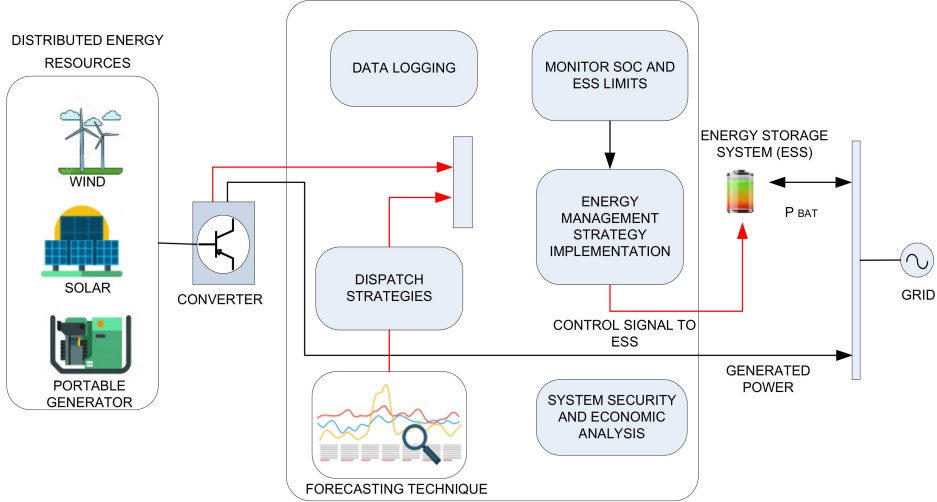

**Figure 8.** Structure of energy management system.

Energy management in microgrids is a complex automated system that is aimed at optimal scheduling of available resources (CG, DGs, ESS) to meet the day-to-day demand while considering the meteorological data and market price. There are three control approaches in energy management of the microgrid which are: (i) centralized, (ii) decentralized, and (iii) distributed.

The centralized control is at the core of the control in this method MGCC, which collects the information from the local controllers and analyzes it to control the system actions [61]. This process requires end-to-end communication between all local controllers to the central controller. Different EMS structures are shown in Figure 9.

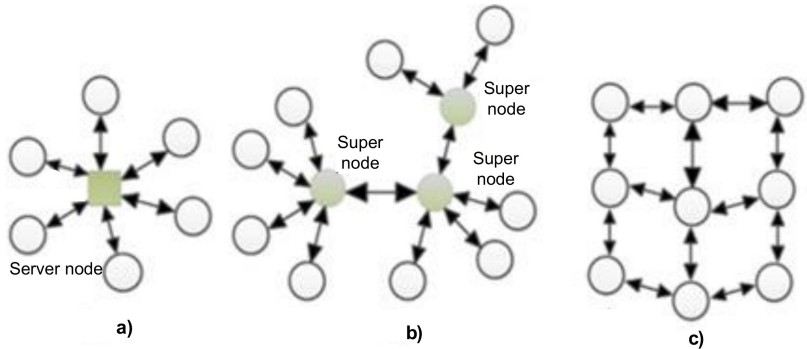

**Figure 9.** Types of EMS control: (**a**) centralized, (**b**) decentralized, (**c**) distributed.

With an increase in the geographical area, the system control in centralized mode becomes difficult due to the delay or lag in the communication, which leads to delay control. This process is not feasible as well as not economical; hence, we choose the decentralized mode of control. In decentralized control, each unit has its own local controller that works in an autonomous state where it receives the voltage and frequency data [62]. Here, the decentralized control does not provide the all the information to the other local controllers, but rather exchanges the global information to make the decisions of the overall system. The exchange of information is allowed in a few controllers to take action spontaneously in a state of emergency. A third approach, obtained with a combination of the above two control approaches, is the distributed control [63]. This mode of control scheme provides control to both centralized as well to decentralized property up to a certain degree of control. In this control scheme, each local controller unit uses the local information like voltage and frequency from the neighbors, which helps to obtain a global solution by the central controller while using the two-way communication link by the local controllers. Characteristics of different types of controls in the energy management system are presented in Table 4.

**Table 4.** Characteristics of different types of controls in the energy management system.

|  | Centralized | Decentralized | Distributed |
|---|---|---|---|
| **Information Accessed** | Microgrids pass information to the central controller | Independent control is provided with data from the other local controllers | Interoperability and data exchange between every device |
| **Communication Information** | Synchronized information from the device to the central controller | Information among local controllers is asynchronized | Communication is both locally and globally asynchronized |
| **Function in real-time** | Complex | Acceptable | Easy |
| **Feature of Plug and play** | The central controller needs to be instructed | Can be accessed by central controller | Available by the peers |
| **Expenditure** | More | Less | Less |
| **Structure of Grid** | Centrally controlled | Locally controlled | Both centrally and locally controlled |
| **Tolerance during fault** | Less tolerance capability | One router fault—tolerated N router fault—expensive | N router fault—tolerated, Possible self-healing feature |
| **Infrastructure** | Needs suggestion integrating DERs | Integration is modular and possible | No change while integration |
| **Size (Number of nodes)** | Less | IPv4-$2^{12}$ IPv6-$2^{128}$ | >$2^{128}$ |
| **Final Nodes** | No identification | Unique identification IP | Global unique identifier |
| **Operation Flexibility** | Very less | Available | Very much needed |
| **Bandwidth & Latencies** | Low and high | Both are great | High and low |
| **QoS** | Not allowed | Allowed | Inherent |
| **Connectivity** | EPA (Physical) | TCP/IP (Physical) | TCP/IP (Virtual) |
| **Safety measures** | Less | Available | High |
| **Individuality** | No | No | Possible |

### 3.2. Data Handling in EMS

Data handling and clustering are the prominent steps towards system management, as many intelligent measuring and sensing devices have been integrated with the MG, which generates a large amount of data per unit (hour, minutes, or seconds). The complex structure of MG system requires it to be equipped with different sensors and monitoring equipment which bring varied kinds of data, like structured data from the conventional power system, semi-structured data from the system like images (camera), unstructured data from meteorological data, network structure, and maps [64]. Figure 10 shows different data types available in EMS.

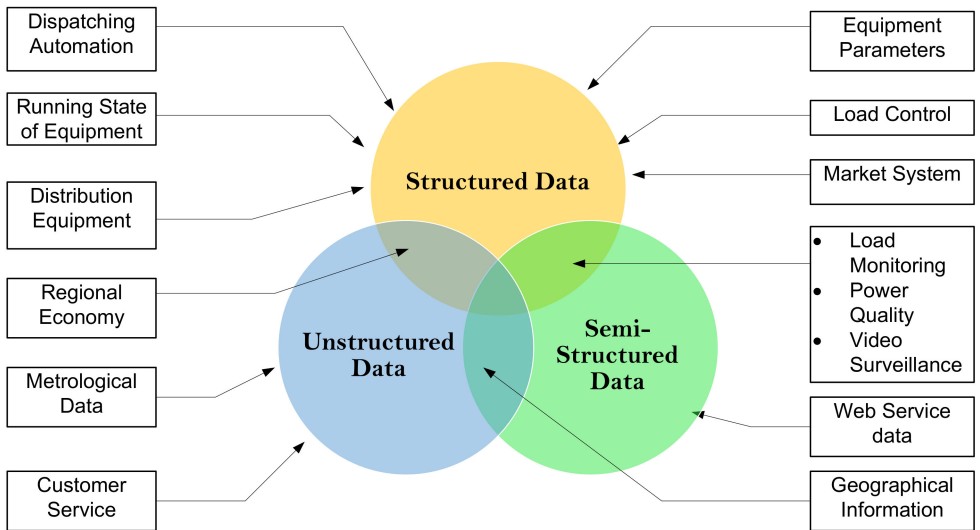

**Figure 10.** Data availability in a microgrid.

Usage of a wide variety of applications of communication and network has improved the speed of the data generated from the units while applications like big data are used to access the information [65]. Intelligent networks help in unfolding the unknown patterns from the data collected. Analytical software technologies like Hadoop, HBase, and Storm are used as data centers to support the vast collection of the data in a structured format by the sensors and the other measuring devices such as smart meters.

### 3.3. Network Reconfiguration

Network reconfiguration is an optimization problem that identifies the optimal radial topology of the distribution network based on all topologies. Network reconfiguration is generally carried out with the aim to reduce the power loss, harmonize voltage profile, and unify network loading through a multi-objective framework. The multi-objective optimal solution problem uses deterministic and stochastic methods for reconfiguration. Much work on reconfiguration is presented using the meta-heuristic method in distribution systems considering radial topologies by interchanging of tie lines [66,67].

### 3.4. Forecasting in EMS

EMS proceeds with data available towards analyzing different forecasting parameters like electricity price market, energy purchase, weather, demand response management, and financial planning using forecasting techniques.

Forecasting is a prominent part of energy management, which is classified in different categories concerning the period of forecast required [68]. These classifications are: (i) very short-term (seconds–$\frac{1}{2}$ h), which is used for the dynamic control of renewable energy sources according to the load requirements; (ii) short-term ($\frac{1}{2}$–6 h), which is used for energy scheduling among the sources and the storage devices; (iii) medium-term (6 h–1 day), which is used for market pricing; and (iv) long-term (1 day–1 week), which is used in load

dispatch and maintenance [69]. Figure 11 shows types of forecasting techniques available in EMS of microgrid.

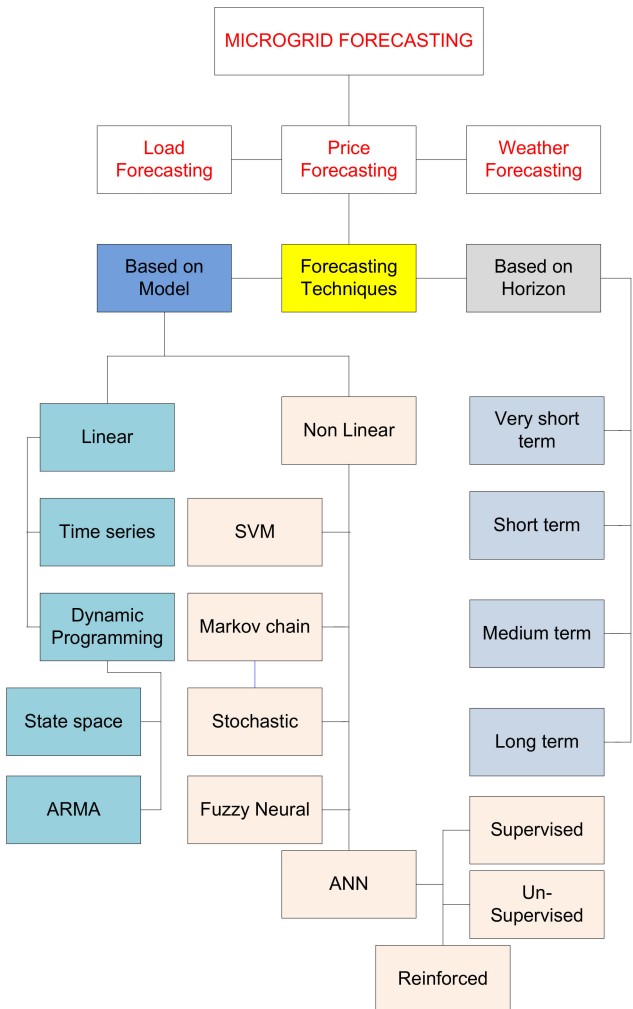

**Figure 11.** Forecasting techniques in EMS.

### 3.5. Demand Management in Microgrid

Load balance acts as a constraint between generation and demand. Load demand balance problems can be categorized in two ways: the supply-side and the demand-side [70]. Supply-side balance can be obtained by using the hierarchical control scheme for the economic scheduling for consumption by the end-users. Load control can be categorized as: (i) controllable loads, which are the loads that are managed according to the price, and (ii) shiftable loads, also known as deferrable loads, such as charging of electric vehicles, washing machines, dryers, which can provide scheduling flexibility for demand response.

The demand-side balance needs to be carefully accessed by modeling the generation in renewable energy, i.e., by forecasting for the supply to the users in the system. Demand-side control is sub-categorized into direct load control (or the demand side management) and price-based load control (or the demand response). Demand-side control is performed by the central controller by the consumer agreement to mainstream the economic agenda. In the price-based load control, the consumer is provided with options to choose their energy consumption according to the market price available. Figure 12 shows different supply and demand classification in EMS.

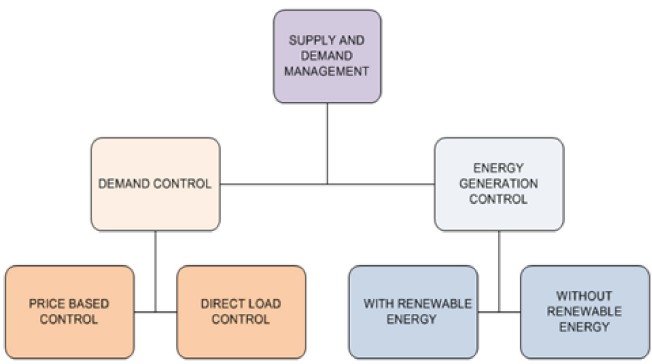

**Figure 12.** Supply and demand management classification.

## 4. Numerical Methodologies of EMS

Different EMS techniques are differentiated according to the numerical methods used for controlling the energy management system. These methods are broadly classified into three categories: (i) classical methods, (ii) metaheuristic methods, and (iii) intelligent methods.

### 4.1. Classical Methods

Classical methods are the mathematical programming or classical programming methods that choose certain variables to maximize or minimize a given function subject to a given set of constraints. Branch and bound are the classic components that are used for solving the classical method approach to find the optimal solution in an iterative process without integer constraints. Classical methods use both linear and nonlinear optimization models to solve the problem. The classical methods are divided into certainty- and uncertainty-constrained problems.

Under certainty linear programming (LP) are mixed integer programming (MIP) and nonlinear programming (NLP). A combination is mixed-integer non-linear (MINLP) and mixed-integer linear programming (MILP) [71–73]. Uncertainty constraints are decision theory (rule-based and deterministic-based), where the output of the model is fully determined by the parameter values and the initial values; whereas probabilistic (or stochastic) models incorporate the randomness in their approach such as dynamic programming (DP) and stochastic programming (SP) [74]. An optimization algorithm is an algorithm that uses the physical deterministic method of solving the solution without any random nature being known as deterministic. Table 5 shows a review of MG EMS by classical methods.

**Table 5.** A review on classical mathematical programming methods used in EMS.

| Ref No. | Method | Power Sources | Ev | Dr | Grid/Island | Ems | Remarks |
|---------|--------|---------------|----|----|-------------|-----|---------|
| [75] | MILP-LP | PV, BT, FC | | | G/I | C | A mixed-mode of EMS is proposed with ON/OFF and continuous run mode. |
| [76] | MILP | PV, WT, BT | | * | I | C | Cost reduced by reducing the ESS with advantageous demand response (DR) determination. |
| [77] | MILP | PV, BT, DE | | * | G/I | C | EMS proposed to minimize the fuel cost while optimizing the diesel generators and battery sizing using a piecewise linear function. |
| [78] | MILP | PV, WT | * | | I | C | Optimizing the day-to-day energy scheduling with DR and EVs using multiobjective constraints. |

**Table 5.** *Cont.*

| Ref No. | Method | Power Sources | Ev | Dr | Grid/Island | Ems | Remarks |
|---|---|---|---|---|---|---|---|
| [79] | MILP | PV, WT, DE, MT, FC, BT | | * | G/I | C | EMS is modeled to optimize while determining the capital cost, cost of the fuel, energy cost, and penalization for emission. Energy sources and storage are considered in economical dispatch for techno-economic analysis. |
| [80] | MILP | PV, BT | | | G/I | C | A three-phase EMS model is proposed with load shedding considering outage constraints. |
| [81] | MINLP | PV, WT, MT, FC, BT | | | I | C | EMS is developed for a three-phase system to minimize the fuel, startup, and shutdown expenditure. |
| [82] | MINLP | PV, BT | | | G/I | C | Stable operation of hybrid MG with clean water supply while reducing the overall daily operating costs. |
| [83] | NLP | PV, FW, MT, FC, BT | | | G/I | C | Energy market operational cost and its profit are determined by the MG management application. |
| [84] | NLP | PV, FC, BT | * | | G/I | C | Maximization of the cost benefiting charge–discharge scheduling of the battery considering the customers' load shifting events. |
| [85] | DP | DE, BT | | | G/I | C | EMS is modeled to optimize the operational cost of the conventional grids considering the penalty cost. Computational time is reduced using Pontryagin's Principle. |
| [86] | DP | WT, DE, BT | | | G/I | C | Minimization of the total cost of operations by scheduling the available units while predicting the wind speed by short-term forecasting and determining the real-time pricing. |
| [87] | Approx. DP | WT, BT | | | G/I | C | Optimization of the MG is proposed considering the cost function of the unit commitment and economic dispatch operations along with daily energy scheduling. |
| [88] | Rule based | PV, UC, MT, BT | | | I | C | To perform power scheduling with day-ahead forecasting for the conventional, PV generators and gas turbine are used in a deterministic power optimization. |
| [89] | Rule-based | PV, WT, BT | | | G/I | C | To configure switches operation to model different configurations considering SOC of the battery and load imbalance. |
| [90] | Rule based | PV, UC, BT | | | G | C | An energy management strategy with PV generator and SOC-based battery hierarchical structure for electricity regulation and continuous operation of the microgrid. |
| [91] | Deterministic based | PV, WT, MT, BT | | * | G/I | C | Proposed to minimize the overall running cost of the system by reducing the industrial loads, considering TOU rate of demand response programs executed. |
| [92] | NP-Hard | PV, BT | | * | G/I | C | Proposes polynomial–time algorithms for approximating optimal solutions and robust supplier networks of group energy communities in terms of a black start while minimizing the operational costs. |

PV—Photo voltaic; WT—Wind Turbine; MT—Micro Turbine; FW—Flywheel; DE—Diesel; FC—Fuel Cell; UC—Ultra Capacitor; G—Grid; I—Islanded; C—Centralized, DC—Decentralized, DT—Distributed, *—Availability.

*4.2. Metaheuristic Methods in EMS*

A metaheuristic is a branch of random search and generation algorithms. These algorithms select a path through a search algorithm such as a heuristic (random) to find an optimal solution in an optimization problem with or without constraints. Metaheuristic algorithms perform computation when incomplete data or limited capacity are provided [93]; the sample set of random values are considered and explored for an optimal solution. Metaheuristic approaches use a separate search strategy to generate a random selection or assumption of the problem variables, which can be advantageous in a variety of situations.

An optimal solution can be found in the distinct search space as used in combinatorial optimization. Metaheuristic method is an iterative method that is unlikely to guarantee a global optimum solution due to its convergence properties. This can be compensated with finding the mean of the solutions; the use of Monte Carlo simulation improves the convergence of the solution. Stochastic implementation of optimization is dependent on the random variables created [94]. The metaheuristic approach works on two concepts, namely intensification and diversification. Intensification is searching a local area to find an optimal solution when we know that solution could be found in the prescribed region. The diversification process is searching the space on a global scale with no limits in the search pattern using the randomly generated variables, while randomization increases the diversity of solution when the search space exceeds the local optima. To find the global optimal or the best solution, both the intensification and diversification processes need to be in proper balance, which increases the rate of convergence in the algorithm [95–99]. A few metaheuristic algorithms are particle swarm optimization (PSO), genetic algorithm (GA), modified PSO (MOPSO), non-dominated sorting genetic algorithm II (NSGA-II), enhanced velocity differential evolutionary PSO (EVDEPSO), priority PSO, multi-voxel pattern analysis (MVPA), grey wolf optimization (GWO), artificial bee colony (ABC), adaptive differential evaluation (ADE), crow search algorithm (CSA), rule-based bat optimization (BO), gravitational search algorithm (GSA), alternating direction method of multipliers (ADMM) using modified firefly algorithm (MFA), teaching–learning optimization (TLA), social spider algorithm (SSO), and whale optimization algorithm (WOA). Table 6 provides a critical review of the metaheuristic methods used in EMS.

**Table 6.** A review of metaheuristic methods used in EMS.

| Ref No | Method | Power Sources | Ev | Dr | Grid/Island | Ems | Remarks |
|--------|--------|---------------|----|----|-------------|-----|---------|
| [100] | NSGA-II | PV, WT, BT | | | G/I | C | A multi-objective optimization problem is proposed to maximize the economy. Intelligent power marketing is adapted to improve the economic dispatch of the microgrid. |
| [101] | NSGA-II | PV, WT, BT | * | | G/I | C | This paper establishes an integral objective function considering the demand response and user satisfaction constraints, which has an effect on the economy and operation of the system with the DR strategy. |
| [102] | PSO | PV, MT, BT, TES | | | G/I | C | An optimal energy planning is proposed for the recently modeled energy hub. An efficient microgrid structure is discussed along with technical and economic prospects with optimization. |
| [103] | CVCPSO | PV, WT, DE | * | | G/I | C | Minimizing the operating costs while maximizing the utility benefit using the CVCPSO algorithm, which yielded the Pareto-optimal set for each objective, and the fuzzy-clustering technique was adopted to find the best compromise solution. |

**Table 6.** *Cont.*

| Ref No | Method | Power Sources | Ev | Dr | Grid/Island | Ems | Remarks |
|---|---|---|---|---|---|---|---|
| [104] | MPVA | PV, WT, MT, BT | | | G/I | C | A sports metaheuristic algorithm to minimize the overall running cost of MG while studying four different MG scenarios. |
| [105] | GWO | PV, WT | | | G/I | C | A sine cosine optimizer is used to optimally participate in the trading of energy, i.e., selling or buying the power while bringing the capital cost of the microgrid. |
| [106] | ABC | PV, WT, DE, BT, FC | * | | G/I | C | An EMS application of the V2G economic dispatch problem is optimized in the MG while converting the multi-objective problem to a single objective using the judgment matrix methodology. |
| [107] | EBC | PV, WT, MT, BT | | | G/I | C | Different TOUs are evaluated to minimize MG operational costs and to analyze the efficiency of a typical distribution system, considering all relevant technical constraints. |
| [108] | ADE | DG, BT | | | G | C | An ADE-based optimization is proposed for the DC microgrid modeling the active power sources under real-time pricing to minimize the total operating cost. |
| [109] | MOPSO | PV, MT, BT, TES | | * | G/I | C | EMS application is proposed to reduce the carbon dioxide emissions and payback period of the microgrid structure. |
| [110] | EVDEPSO | PV, BT | * | * | G/I | C | A day-ahead planning schedule is determined to improve the energy market trading while managing the resources available. Includes the electric vehicles participating in the energy market, G2V and V2G. |
| [111] | Rule base BO | PV, WT, MT, FC, BT | | * | G | C | A bat algorithm is used to optimize the MG operation by forecasting the load power and uncertainties in RES using probabilistic methods. The weight factors are taken for tuning. |
| [112] | CSA | PV, FC, DE, HY | | * | G/I | C | The Pareto front is considered to investigate the operating cost, solar power uncertainty, carbon emission, and the cost of the parameters. Hydrogen fuel is considered in reducing operating costs. |
| [113] | GSA | PV, WT, BT | * | * | G/I | C | Optimization of the overall cost considering the carbon emission and weekly generation scheduling for the small dispatchable systems. |
| [114] | ADMM-MFA | PV, WT, MT, FC, BT | | * | G/I | C | EMS is modeled for the MG to optimize the electricity price by considering the load profile, PV irradiance, and market prices with certain constraints. |
| [115] | TLA | PV, WT, MT, FC | * | * | G/I | C | Hybrid MG reducing the operating cost considering thermal power recovery and hydrogen generation; V2G technology helps to convert the PEVs into active storage. |
| [116] | SSO | PV, WT, DE, FC | | * | I | C | Optimal sizing of the renewable energy sources with conventional sources to minimize the cost of energy (COE) and power loss supply probability while analyzing the reliability. |
| [117] | WOA | PV, WT, DE, BT | | * | I | C | EMS is proposed to optimize the load demand of the MG by minimizing the operating cost with improved reliability of the power. |

PV—Photo voltaic; WT—Wind Turbine; MT—Micro Turbine; TES—Thermal energy storage; DE—Diesel; FC—Fuel Cell; HY—Hydro; C—Centralized, DC—Decentralized, DT—Distributed, *—Availability.

*4.3. Intelligent Methods in EMS*

4.3.1. Fuzzy Control and Neural Networks

For the computation of a large amount of numerical processing data like signals or images, fuzzy logic systems and neural networks (NN) are used. These methods are computational nonlinear algorithms with the flexibility to use a range from small software programs to large hardware systems. Through continuous decision-making by the system, learning takes place and the knowledge acquired is stored in as weights. These weights are the internal parameters of knowledge. A fuzzy logic system, when used to control a system through a set of rules considering the constraints, is known as fuzzy logic control (FLC). Applications of FLC are used to improve battery state of charge (SOC), smooth voltage profile, and grid-to-vehicle (G2V) charge transfer [118].

Neuro-fuzzy is a combination of fuzzy approach and neural network, where fuzzy inference system (FIS) is adjusted by the data provided to NN learning rules. Improved speed, accuracy, and strong learning skills along with simple execution are the advantages of this approach [119].

A neural network is an interconnection of neurons, when used in a physical system to control using different layers of connection, which are also known as an artificial neural network (ANN). These artificial NN are used for adaptive control and model predictive analysis. Applications of ANN are provided with training via a dataset. From experience or the outputs of the model, self-learning takes place. Using ANN for MG, EMS can perform complex operations such as forecasting DR and control of MG [120].

Recurrent neural network (RNN) is a classification in ANN which allows it to provide temporal dynamic behavior and the structure of RNN connects the temporal sequence through the graph between the nodes. Similar to feed forward neural networks or ANN, which process variable-length sequences using internal memory, RNN has an internal state memory to process the sequence of inputs using short-term memory (STM) or long short-term memory (LSTM) for predictions of energy and economy. A review on fuzzy and ANN-based applications in EMS are described in Table 7.

**Table 7.** A review on fuzzy and ANN-based applications in EMS.

| Ref No. | Method | Power Sources | Ev | Dr | Grid/Island | Ems | Remarks |
|---|---|---|---|---|---|---|---|
| [121] | Fuzzy logic | PV, WT, BT | | * | G/I | C | EMS for distributed generations DGs in AC MG. An adaptive neuro-fuzzy inference system (ANFIS) is developed to manage the available energy in ACMG. |
| [122] | Fuzzy | PV, WT, FC, BT | | | I | C | The system is controlled by a low complexity fuzzy system, with only 25 base rules which give better results in terms of control and energy-saving efficiency, that has been improved. |
| [123] | Fuzzy logic | PV, WT, DE, BT | * | * | G/I | C | Studies different fuzzy techniques for the charging/discharging of the electric vehicle while ensuring the optimal demand management from the vehicle-to-grid (V2G). |
| [124] | Fuzzy | PV, FC, BT | | * | G/I | C | EMS is developed to manage the operating conditions with economic constraints. Operations of grid ON/OFF connections are also discussed using the fuzzy logic controller and a predictive controller. |

**Table 7.** *Cont.*

| Ref No. | Method | Power Sources | Ev | Dr | Grid/Island | Ems | Remarks |
|---------|--------|---------------|-----|-----|-------------|-----|---------|
| [125] | FLC | PV, BT | | * | G/I | C | A fuzzy logic-based energy management system is developed to minimize the power-sharing error between renewable energy sources and demand. |
| [126] | Neuro-fuzzy | PV, WT, MT, FC, BT | * | | G/I | C | A neuro-fuzzy Laguerre wavelet control (FRNF-Lag-WC) architecture scheme is validated for various stability, quality, and reliability factors obtained through a simulation testbed implemented. |
| [127] | Neuro-fuzzy | PV, FC, BT | | * | I | C | A battery cycle is improved by reducing the charging/discharging period and ensuring optimal power-sharing in the microgrid. |
| [128] | RNN. | PV, BT | | * | I | C | A control strategy is developed to maximize consumption and minimize electricity pricing by using an LSTM forecasting method for supply–demand management. |
| [129] | ANN | PV, WT, DE, BT | | * | I | C | A real-time scheduling problem is developed for an MG with a finite horizon model using the ADP approach. The ADP approach is modeled using the RNN technique. |
| [130] | RNN | PV, BT | | * | I | C | Discussed many algorithms for scheduling including the maximum time lap scheduling and day-ahead forecasting for a building of its energy consumption with PV installation. |
| [131] | ANN | PV, WT, MT, DE, BT | | * | G | C | EMS application to optimize the economic dispatch and to minimize the operating cost in a hybrid microgrid using Lagrange programming neural network. |

PV—Photo voltaic; WT—Wind Turbine; MT—Micro Turbine; FW—Flywheel; DE—Diesel; FC—Fuel Cell; UC—Ultra Capacitor; C—Centralized, DC—Decentralized, DT—Distributed, *—Availability.

## 4.3.2. Model Predictive and Multi-Agent EMS

Model predictive control (MPC) is an algorithm that regulates or controls the system based on the moving or rolling horizon approach as specified in Unnikrishnan et al. [129]. The role of MPC is to make the system less sensitive to the variables and control the physical process. MPC can be performed online with uncertainty constraints. In online methods, the current system parameter and forecasted parameters help in updating the decision variables at any instant [132,133]. The optimum solution could be obtained by updating decision variables with current system parameters with ease, and gets complex with an increase in variables. Hence, it is used in smaller systems.

In the multi-agent system (MAS), the objectives of the system are obtained by intelligent agents communicating with other nearby agents while participating to form a configuration. MAS is an online/offline approach used in MG applications as shown in [134]; this approach is utilized in the control of EMS, optimization, and managing of the energy market. In [135,136], the application of MAS is used to control the architecture of the MG while using optimization techniques for the configuration of renewable resources. Table 8 presents the review on MPC and MAS based on EMS.

<div align="center">**Table 8.** A review on MPC and MAS based on EMS.</div>

| Ref No. | Method | Power Sources | Ev | Dr | Grid/Island | Ems | Remarks |
|---|---|---|---|---|---|---|---|
| [137] | MPC | PV, FC, SC, DE, BT | | * | I | DT | Energy scheduling is proposed using the MPC to optimize the dwell time of the high SoC state of the battery and to smoothen the set point deviation of the fuel cell for regenerative capability. Compared with fuzzy-based heuristic in generation and load demand. |
| [138] | MPC | PV, WT, BT | | * | G/I | DT | MPC-based decision-making is developed by the optimization algorithm for participation in the grid electricity market with excess generation to support ancillary services of the main grid. |
| [139] | MPC | PV, BT | | * | G/I | DT | A real-time microgrid from Athens is developed in the laboratory to study the day-ahead market and the control management of the energy profile with the energy market. User interface with the market interactions is performed for an enhanced microgrid. |
| [140] | Adaptive MPC | PV, DE, BT | | * | I | DC | An EMS application is developed to optimize the cost function of the fuel in a diesel generator for economic dispatch using the Lagrange multiplier and lambda iteration method with battery operation constraints. |
| [141] | MPC | PV, BT | * | * | G/I | DT | An MPC-based control strategy is developed to sell or store the excess generated power from the solar panels while managing the overall conditions like heating, ventilation, air conditioning system, time of use pricing, and to reduce economic constraints. |
| [142] | MPC | PV, BT | * | | G | DC | By installing an ESS at the end of the feeder, the capacity of PVs and EV connected to the bus are extended up to twice the capacity of the main power source. |
| [143] | MPC | PV, WT, DE, BT | | * | G/I | DC | A proximate scenario is taken by the optimizer at each step, and the optimal supply of system capacity is accessed based on the scenario selected and the possible variations in the future. |
| [144] | MAS | PV, WT, MT, FC, BT, DE | | * | I | DC | MAS-based agent optimization is developed to optimize the operation of the distribution system with DG in energy scheduling and generation. EMS is performed for the system by considering the constraints, such as generation cost and emission of carbon. |
| [145] | MAS | PV, BT | * | * | I | DC | A MAS-based two-stage energy management system is developed using the Kantorovich method for the energy generation scenario considering the self-healing strategy by the decentralized restoration technique and coordinated management. |
| [146] | MAS-CNN | PV, WT, DE, BT | | * | G/I | DC | MAS-based energy management is proposed for the generation management of the PV, wind, and load. Balancing is maintained using the CNN (convolution neural network)-based load forecasting technique for the load demand. |
| [147] | MAS | PV, DE, BT | * | * | I | DC | This paper proposes a MAS-based intelligent energy management system to operate a hybrid microgrid in islanding mode while effectively minimizing the peak demand of the system using the V2G and LED savings. |
| [148] | MAS | PV, WT, FC, BT | | | G/I | DC | This paper proposes a communication rule for sharing the local information of the agents and getting access to the global information was based on an average consensus algorithm (ACA), and a restoration decisions strategy based on the discovered global information was developed. |
| [149] | MAS-RL | PV, WT, BT | | * | G/I | DC | A multi-agent-based EMS is developed to manage the objectives of the system. Reinforced learning is imbibed with MAS to improve the decision-making capability by learning using the sets for the participation in the energy trade marketing. |
| [150] | MAS | PV, WT, BT | | * | G/I | DC | Experimental results show the ability of the proposed multiagent T-Cell-based RT-EMS in maintaining the stability and smooth operation of the MG with modularity and fault tolerance features implemented through the MAS JADE platform. |

PV—Photo voltaic; WT—Wind Turbine; MT—Micro Turbine; DE—Diesel; FC—Fuel Cell; SC—Supercapacitor; G—Grid; I—Islanded; C—Centralized, DC—Decentralized, DT—Distributed, *—Availability.

### 4.3.3. Game Theory and Deep Learning

Deep reinforcement learning (DRL) is an intelligent algorithm approach to solve complex problems like decision-making through training or learning. It is a combination of reinforced learning (RL) and deep learning (DL) where agents perform the decision-making task to a wide variety of applications. DRL is a sub-category of intelligent machine learning, which is also a part of artificial intelligence where a system learns from the actions it performs as a human learning experience [151,152]. The agent learns by a reward and penalty system on their decision policy.

Game theory (GT) brings multiple decision variables to interact using a mathematical model to analyze the environment. The objectives of the problem are achieved by introducing each strategic decision-making variable to participate in the game. Nash equilibrium is a prominent solution concept for game theory, where the actions of other players are set to constant while there is no change to the unilateral strategy by any player to change their revenue strategy. Thereby, it is possible to arrive at an optimum mutual response from all the players [153]. To find the optimal solution in the non-cooperative game theory when there is evidence that no leader–follower relationship is found, Nash equilibrium strategy is used to improve the utility parameter by making every player compete against each other. A review on game theory and deep reinforced learning in EMS has been presented in Table 9.

**Table 9.** A review on game theory and deep reinforced learning in EMS.

| Ref No. | Method | Power Sources | Ev | Dr | Grid/Island | Ems | Remarks |
|---------|--------|---------------|----|----|-------------|-----|---------|
| [154] | DRL | WT, DE, BT | | * | I | DC | An EMS is proposed for energy storage management and load shedding management with dual control policy to manage the utility of the system dual control to improve resilience. The dual controls are the energy storage and load shedding policies. |
| [155] | DRL | BT | * | | G | DC | EMS is developed to manage fuel efficiency compared to the rule-based approach. The EMS developed makes decisions by itself from the actions of the states. |
| [156] | DRL | PV, WT, BT | | * | I | DC | DRL-based energy management is proposed to minimize the operating cost and to improve the economic performance of the islanded microgrid by controlling the energy reserve. |
| [157] | DRL | PV, WT, MT, FC, BT | | * | G/I | DC | An EMS is modeled with DRL and the Markov decision process (MDP) strategy to satisfy the objective function, i.e., by minimizing the overall operating cost of the MG system. |
| [158] | RL | WT, BT | | * | G/I | C | An EMS application for the consumer-based intelligent method is developed for the consumer to explore and control the stochastic nature of the generation and load actions. |
| [159] | DRL | PV, WT, MT, FC, BT | | | G/I | DC | Paper proposes a scheduled strategy to minimize the daily operating cost of the MG using DRL architecture for addressing the problem of operating an electricity MG in a stochastic environment. |
| [160] | Game Theory | PV, WT | | | G | C | A game-theory-based EMS is modeled to minimize the utilization cost of the system using the coalition theory, the EMS is proposed to reduce the utilization cost while improving the market profit of the sellers. |
| [161] | Game Theory | PV, WT, BT, HYD | * | * | G/I | DC | A Nash equilibrium-based game theory EMS is modeled for controlling the power exchange and minimizing the operating cost. An optimal operation can be achieved by maximizing the preferences of the agents using the Nash equilibrium. |
| [162] | Game Theory | PV, BT | | * | G/I | DC | An MG-based non-cooperative game theory EMS is modeled to optimally decide the electricity price for the consumers by regulating the storage capacity of the system. A mechanism for the price regulation is developed for the modeled EMS. |
| [163] | Game Theory | PV, BT | | * | I | DC | Optimal scheduling of the energy and storage management is proposed by the continuous non-cooperative game-theory-based energy management system by considering the energy consumption scenario to reduce the overall cost. |

<div align="center">

**Table 9.** *Cont.*

</div>

| Ref No. | Method | Power Sources | Ev | Dr | Grid/Island | Ems | Remarks |
|---------|--------|---------------|----|----|-------------|-----|---------|
| [164] | Game Theory | PV, WT, BT | * | * | I | DC | An EMS is developed by forecasting the generation of the short-term wind power plant using big data. The optimal payment period is decreased by finding the prediction error of the MG. |
| [165] | Game Theory | PV, WT, FC, BT | * | | G/I | DT | The paper gives cooperation between the agents as a non-cooperative or a cooperative game theory approach. Nash equilibrium is used for exploring the optimum solutions of games with energy management. |

PV—Photo voltaic; WT—Wind Turbine; MT—Micro Turbine; FW—Flywheel; DE—Diesel; FC—Fuel Cell; UC—Ultra Capacitor; G—Grid; I—Islanded; C—Centralized, DC—Decentralized, DT—Distributed, *—Availability.

### 4.4. Problem-Based Classification

The microgrid energy management strategies are discussed in previous sections, and objectives considered in the review can be further classified into problems addressed. The review methodologies that are classified based on problems addressed are shown in Table 10.

<div align="center">

**Table 10.** The problem addressed in microgrid energy management.

</div>

| Problems Addressed | References |
|--------------------|------------|
| Optimal storage management | [76,98,112,123] |
| Demand response program | [77,92,95,151,163] |
| On vehicle-to-grid system (V2G) | [78,108,113,118,124] |
| Cost minimization | [79,81,82,84,91,93,94,99,100,105,106,110,111,127,141,151,152,161,163] |
| Energy schedulling | [80,87,89,90,102,104,107,109,114,115,121,136,139,142,154,156,157,160] |
| Operating time | [85,126,150] |
| Reliability of operation | [116,117,143,165] |
| Communication and information exchange | [129,134,140] |
| Based on forecasting | [119,129,138,146,162,164] |
| Data collection and scenario generation | [125,147,155,158,159] |
| Based on market participation | [83,88,101,132,137,141,146,149,153] |
| Time response | [96,97,128,130,131] |
| Stability analysis | [86,120,136,145,148,150] |
| Generate energy with lower emissions | [103,144] |

## 5. Microgrid Standards

Standards are the parameters or the process which ensure the product's performance levels to satisfy the safety and quality for the implementation according to utility market requirements. The standards are developed to set a standard in the market for the safety of consumers [166,167], introducing a set of verification procedures to test the performance of the quantification and their comparison with a minimum set of requirements. Standards for microgrids are set to provide configuration, topology, and laws to control the microgrid and its integration to renewable sources. Different configurations can be implemented with microgrid blocks to perform different operations. A set of testing procedures is carried out in the distributed network operator [168] (DNO) and microgrid operator with parameters to compare their control functions. These metrics or parameters are designed to test the endurance of the system. Standards that exist for the smart grid distribution network are the Institute of Electrical and Electronics Engineers (IEEE 1547) with identification code 1547, which provides guidelines for interconnecting dispatchable sources into the electric power grid; and IEEE 2030, which provides the inter-operability guide between smart grids and microgrids [169]. International Electro-Technical Commission (IEC) is another standardization for microgrids in which IEC 62,898 provides design and implementation of the microgrid. For electric vehicles in IEEE 2030.1, IEC 61851, and ISO 15118-1 give the guidelines for electric transportation and its interconnection to the power system.

IEEE 1646 and IEC 61850-7-420 provide the standards of communication in the electric network. IEEE 2413 and IEC 61,968 give the standards for connecting IoT into the system and data exchange between devices and the network, respectively. Table 11 presents the standards for microgrid and electric vehicles.

**Table 11.** Standards for microgrids and electric vehicles.

| Standards | Description |
|---|---|
| IEEE 1547 | The standard for interconnecting distributed resources with an electric power system |
| IEEE 1547.1 | Test procedures for equipment interconnecting distributed resources |
| IEEE 1547.2 | Application guide for IEEE 1547 for interconnecting distributed resources |
| IEEE 1547.3 | Monitoring, information exchange, control of distributed resources |
| IEEE 1547.4 | Design operation and integration of distributed resources |
| IEEE 1547.6 | Interconnecting of distributed resources for distribution system secondary networks |
| IEEE 1547.7 | Guide to conducting distribution impact studies for distributed resources interconnection |
| IEEE 1547.8 | The practices identified in P1547.8 should lead to the development of advanced hardware and software and help streamline their implementation acceptance, resulting in higher penetration levels of DER |
| IEEE 2030 | Guide for smart grid interoperability |
| IEEE 2030.1 | Guide for electric power sourced transport infrastructure |
| IEEE 2030.2 | Guide for interoperability of energy storage systems integrated with electric power infrastructure |
| IEEE 2030.3 | The standard for test procedures of energy storage systems integrated with electric power applications |
| IEEE 2030.4 | Guide for control and automation installations applied to the electric power infrastructure |
| IEEE 2030.5 | The standard for smart energy profile 2.0 application protocol |
| IEEE 2030.6 | Guide for the benefit evaluation of electric power grid customer demand response |
| IEEE 2030.7 | The standard for the specification of microgrid controllers |
| IEEE 2030.8 | Standard testing of microgrid controllers |
| IEEE 2030.9 | Recommended practices for the planning and design of the microgrid |
| IEEE 1646 | Communication requirements in substation |
| IEEE 2413 | The standard for an architectural framework for the Internet of Things |
| IEC 62898-1 | Guidelines for planning and design of microgrids |
| IEC 62898-2 | Technical requirements for operation and control of microgrids |
| IEC 62898-3-1 | Technical requirements for the protection of microgrids |
| IEC 62898-3-2 | Technical requirements of microgrid EMS |
| IEC 62898-3-3 | Technical requirements of self-regulation of dispatchable loads in microgrids |
| IEC 62257-9-2 | Recommendations for renewable energy and hybrid systems for rural electrification—Part 9-2: Integrated systems—Microgrids |
| IEC 61850-7-420 | Communication between devices in transmission, distribution, and substation automation |
| IEC 61968 | Data exchange between devices and networks in the power distribution domain |
| IEC 61851-1 | Electric vehicle on-board charger EMC requirements for conductive connection to AC/DC supply |
| IEC 61851-23 | DC electric vehicle charging station |
| IEC 61851-24 | Digital communication between a DC EV charging station and an electric vehicle for control of DC charging |
| ISO 15118-1 | Vehicle-to-grid communication interface—Part 1: General information and use-case definition |
| ISO 15118-2 | Network and application protocol requirements |
| ISO 15118-3 | Physical and data link layer requirements |
| ISO 15118-4 | Network and application protocol conformance test |
| ISO 15118-8 | Physical layer and data link layer requirements for wireless communication |

## 6. Auxiliary Infrastructure

In order to make a smart distribution system operable, a complex of networks and devices needs to get together for a reliable system. IoT and smart meters technologies are the primary components to make the conventional connection between the prosumer and operator into a smart interdependent system with faster and reliable communication [170].

### 6.1. IoT Sensors

Advancements in wireless technology with improved sensing devices using embedded processing technology have led to the Internet of Things [171], which provides efficient monitoring, measuring, and control services.

IoT connects the physical and digital components without any mediation of the operator. The connection of each network device is possible through foolproof protocols. Unique identifier (UID) is a unique identification number for each IoT device that makes it recognizable to others or the control network.

According to Gartner, the number of IoT devices in use by the year 2020 is estimated to be 20 billion. Figure 13 shows the graph of the rate of increase in IoT devices by the year. IoT devices are used in health care sectors (popular IoTs are fitness band and health monitoring devices), the industrial sector (sensing and measuring devices), security sector (cameras and positioning systems), and general devices are used in smart homes for the monitoring and control of loads. Microgrids come into this cross-industry sector: this sector specifies special devices that improve the efficiency of other network devices that include improvements in quality of monitoring and reducing the losses through effective control of failure rate in production [172]. Figure 14 shows the IoT based support to the microgrid applications.

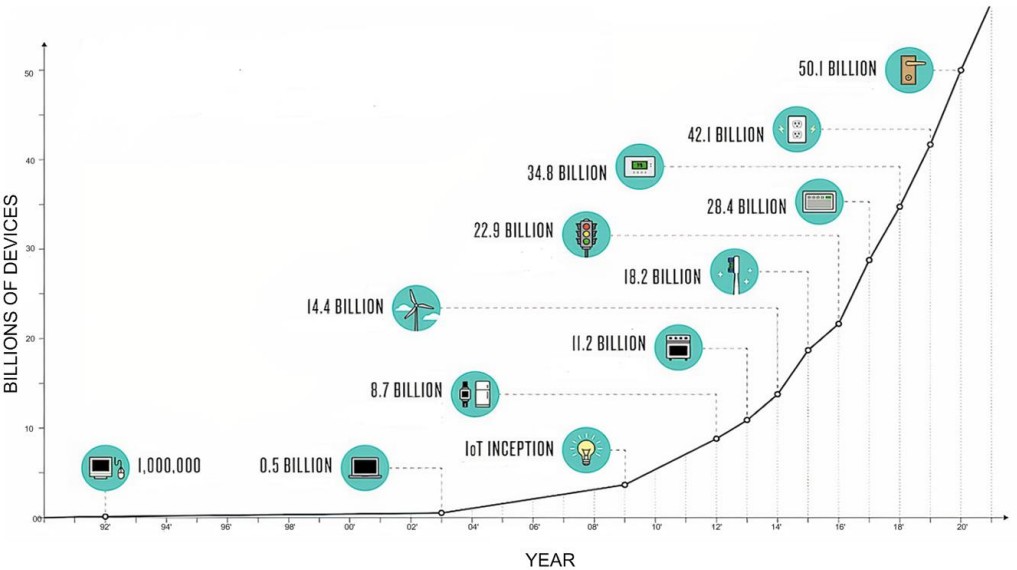

**Figure 13.** Internet of things (IoT) rate of increase in usage in different applications.

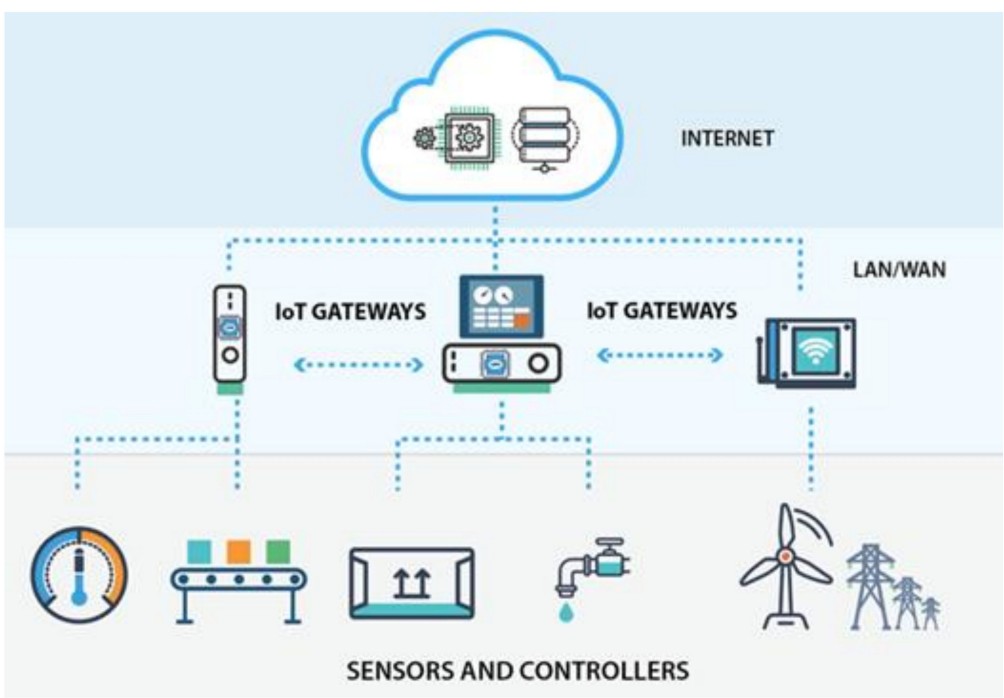

**Figure 14.** IoT support to the microgrid.

### 6.2. Smart Meters

For the last few years, disc-type meters have been replaced by electronic integrated circuit embedded meters which are used effectively by the distribution utility companies in providing authentic and electronic billing for the customers [173]. The necessity for refined flexible billing and control of billing information for two-way power flow proposes the implementation of smart meter technology. Smart meter technology provides the day-to-day of market prices of the power demand to the customer in commercial situations and industries. Previously existing automated meter reading (AMR) technology collects the energy consumption data from the customers to the utility, which is a one-way flow in power and communication. The AMR, an advanced metering infrastructure (AMI) developed in recent years, provides two-way communication and power flow between the meter and the central control system [174]. The improved functionality characteristics from the AMR to the AMI are shown in Figure 15.

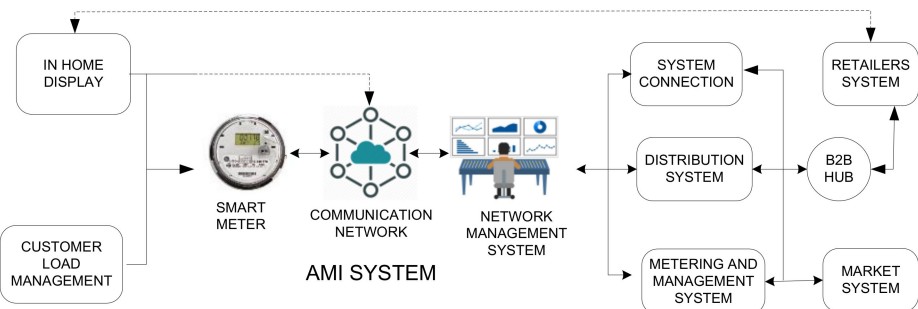

**Figure 15.** AMR and their functionalities in the microgrid.

In the aspect of both transmission and distribution, a smart grid is a revolutionary approach and the smart meters play a significant role as an integral part of the smart grid in communicating with the customers and data collection. Supposedly, the smart meter consists of three main components, which are communication network management, advanced metering element, and data management unit. The smart meter is equipped with a memory device that allows consumers to monitor their energy usage via a software

interface, allowing it to communicate in two ways. The smart meter controls the operation distribution system switches and reclosers which provide an efficient delivery system and maintain reliability. The availability of two-way communication and the energy interface in the smart meter allows the control of distribution infrastructure by sending commands to the control center, which is also known as the distribution automation at the load end. The advantage of the smart meter is that it enables the central control to take action when tampering happens with the available rapid report sent from the smart meter as a part of collecting data [175]. This helps in reducing power theft while improving the power system security. Availability of day-to-day billing reports to the consumers helps them to manage the loads and reduce their bills through the smart meter.

The data from every meter can be collected, processed, and stored using applications like big data [176]. This makes the utility companies go towards the implementation of smart meters where two-way communication plays a prominent role.

## 7. Conclusions

This paper gives a detailed review of the recent analysis of the different energy management strategies proposed for the microgrid, consisting of classical, heuristic, and intelligent algorithms. Furthermore, this paper provides a brief introduction about the architecture of microgrids, different classifications in microgrids, components of a microgrid, communication technologies used, standards available for the implementation, and auxiliary services required in the microgrid. It discusses key applications in energy management, which include forecasting, demand response, data handling, and the control structure. This article also presents an insight on areas in which the scope of research and their contribution to energy management is in the nascent stage.

Optimization in cost minimization, operation control, reliability, energy scheduling, emission control, and load forecasting is the objective functions of the EMS in both the modes of microgrid operation for sustainable development. This makes the MG energy management a multi-objective optimization problem considering the economic, technical, and emission aspects as key constraints. The prime aspects that are covered in this review are on prospects, solutions, and opportunities of the objective functions of the EMS using efficient strategies. Based on the practicability, suitability, and tractability of the methods, the techniques are considered to find global solutions to the operations of the system. The microgrid energy management objectives depend on its mode of operation, whether it is centralized, decentralized, or distributed operation, several economical constraints, and the dynamic nature of dispatchable energy sources. Furthermore, few authors have considered greenhouse gas emissions as an additional objective function apart from non-renewable generators, batteries' health status, integration of active demand response, active and reactive losses along with resilience and customer management.

Many research articles have been published on the energy management of microgrids on different applications, yet the reviewed papers have been considered based on diversity of the objective functions. The areas such as customer confidentiality regulations, management of communication systems, and reliability studies on islanded mode have further scope to emphasize in future studies. Potential areas as mentioned above needed to be focused in detail along with the depth of discharge of the batteries, effect of the conventional grid on greenhouse gas emissions, and demand response integration to obtain effective and efficient operation of microgrids.

**Author Contributions:** Conceptualization, A.R.B. and S.V.; methodology, S.R.S.; software, A.R.B.; validation, A.R.B., S.V. and S.R.S.; formal analysis, A.R.B.; investigation, S.V.; resources, S.R.S.; data curation, A.R.B.; writing—original draft preparation, A.R.B.; writing—review and editing, A.R.B., S.V. and S.R.S.; visualization, A.R.B.; supervision, S.V. and S.R.S.; project administration, A.R.B.; funding acquisition, S.V. and S.R.S. All authors have read and agreed to the published version of the manuscript.

**Funding:** This research was funded by National Institute of Technology Andhra Pradesh (NIT-AP) and Woosong University's Academic Research Funding–2021.

**Acknowledgments:** We acknowledge National Institute of Technology Andhra Pradesh (NIT-AP), Tadepalligudem, Andhra Pradesh, India; and Woosong University, South Korea for their support in carrying out this research work.

**Conflicts of Interest:** The authors declare no conflict of interest.

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
