# Peer review of "Review of Energy Management System Approaches in Microgrids"

_energies, doi:10.3390/en14175459_

Round 1
Reviewer 1 Report
This paper proposes a survey of the Energy Management approaches for microgrids. In general, the paper is interesting, even though several clarifications should be added.
All the following indicated aspects should be clarified and better explained in the manuscript.
Introduction
- The authors should better highlight the innovative aspects of their survey in the manuscript.
Definitions
- A formal definition of microgrid at the beginning of Section 2 is missing. Do the authors consider electric microgrids or even multi-carrier microgrids?
- The list of the main components of microgrids is not complete. For instance, (smart) loads are not included.
- Several recent scientific studies on energy scheduling, show that loads could be classified into non-controllable, shiftable and controllable comfort-based loads and controllable energy based loads. The Authors should comment this point.
- Scarabaggio et al., "Distributed Demand Side Management With Stochastic Wind Power Forecasting," in IEEE Transactions on Control Systems Technology, doi: 10.1109/TCST.2021.3056751.
- Sortomme et al. "Optimal power flow for a system of microgrids with controllable loads and battery storage." 2009 IEEE/PES Power Systems Conference and Exposition. pp 1-5.
(documents that could be cited in the text)
Survey methodology
- The description of the methodologies employed to search for the studies discussed in the review is missing. The authors could add some comments about this point.
Review
- The Energy Management and Power Management approaches could be generally classified into:
- planning methodologies, used for strategic/tactical purposes;
- control techniques, used for operational purposes.
It is not clear if the authors address both these aspects in the survey.
- Do the authors consider the problem of microgrid reconfiguration? This is one of most addressed problem in the planning phase of microgrids. The Authors should comment this point.
-
- M. Helmi et al., "Efficient and Sustainable Reconfiguration of Distribution Networks via Metaheuristic Optimization," in IEEE Transactions on Automation Science and Engineering, May 2021.
- A. Muhammad et al., "Distribution Network Planning Enhancement via Network Reconfiguration and DG Integration Using Dataset Approach and Water Cycle Algorithm," in Journal of Modern Power Systems and Clean Energy, vol. 8, no. 1, pp. 86-93, January 2020.
-
(documents that could be cited in the text)
- The energy management approaches usually are classified into deterministic (considering no uncertainty) and robust/stochastic (dealing with uncertainty). The Authors should comment this point.
-
- Nassourou, M. et al. Robust Economic Model Predictive Control Based on a Zonotope and Local Feedback Controller for Energy Dispatch in Smart-Grids Considering Demand Uncertainty. Energies 2020, 13, 696.
- Karimi, H., and Jadid, S. (2020). Optimal energy management for multi-microgrid considering demand response programs: A stochastic multi-objective framework. Energy, 195, 116992.
-
(documents that could be cited in the text)
- It seems that the authors classifies the surveyed works in accordance with the employed methodologies. In the discussion part, it could useful to add a table containing a classification of the surveyed works in accordance with the type of the addressed problem.
Minor
- Mainly the English is good and there are only a few typos. However the paper should be carefully rechecked.
- The authors should check that all the used acronyms are explained the first time they are used (e.g., in the abstract MG, CG, DG are not defined).
Reviewer 2 Report
In this review article, the authors provided discussions on different numerical-based energy management techniques that are proposed for microgrid management. However, this reviewer has the following concerns on the submitted document:
- The title of the article should be changed from ‘A Comprehensive Review of Energy Management System Approaches in Microgrids’ to ‘Review of Energy Management System Approaches in Microgrids’ as all relevant references cannot be reviewed in one article comprehensively.
- In the abstract section, the authors have wasted 90% effort discussing less important things. They have just mentioned their contributions in one sentence. Therefore, this reviewer suggests the authors to completely revise the section where the background of the study can be completed in 1 to 2 sentences. Then, they should outline the methodology. Finally, they can discuss their contributions and impact of this kind of article.
- The article is full of grammatical mistakes and typos. For instance, they have capitalized the first alphabet of a few words and for the remaining words they have used ‘small letters’ in the keywords section. There are so many similar issues throughout the manuscript.
- Many sentences should be rephrased and restructured.
- The authors should remove the less important sentences and phrases from the manuscript as it become very wordy and lengthy.
- Texts in many figures are too small and very hard to read. Therefore, this reviewer recommends the authors to use unified fonts and their sizes throughout the manuscript. Also, it is highly recommended to use high-resolution images. Last but not least of the figures, they have forgotten to add figure number 15 in the manuscript. It seems the authors are so busy even to check these types of issues.
- They have started the conclusions section with the statement: ‘A microgrid consists of renewable resources, distribution generations, microgrid central controller, demand response, communication devices, load forecasting, and local controllers.’ It is really very weird to see somebody is defining ‘microgrids’ in the conclusions section in a research article. Please read the authors' guidelines and go through some generic YouTube videos regarding the guidelines for research article writing.
- Besides, why did the authors need to provide abbreviations of phrases in the conclusion sections that were not used anywhere in the article!
- Please add a section to discuss your findings and provide recommendations for the researchers and the decision-makers.
- In the tables, please include the drawbacks of the discussed methods (if any).
Overall, the article should not be accepted in its present form. It requires extensive revision and research to get accepted for high-quality journals like 'Energies of MDPI.'
Reviewer 3 Report
This paper, although reasonably well organized, is simply a re-statement of what is well known to the experts working in the field.
There are no critical insights and no weighing of the evidence to guide a reader to any actionable recommendation. It simply describes various component parts & says what they are. Nothing inherently wrong with what is described: but my open question is: and then what?
This paper is a simple description of what others have said in the literature.. I am not convinced it is an advancement towards the implementation of any advanced micro-grid strategy.
I would encourage the authors to develop a specific use case or a number of examples to illustrate quantitatively the application of fuzzy methods, or game theory or integration of EVS and storage in a specific community or for use to support an industry/commercial enterprise.
The paper would benefit from a focus on a use case illustrating the various concepts described.
Round 2
Reviewer 1 Report
Previous comments and concerns have been sufficiently addressed. In the revised paper several improvements have been added.
Reviewer 2 Report
This reviewer would like to thank the authors for revising their article, considering the reviewers' concerns, and submitting it again to the ‘Energies of MDPI Journal’. He believes and feels that the revision made the manuscript more readable and enhanced the overall quality. The revised version of the document is technically sound, well organized, and well written. Also, the authors have addressed significant concerns raised by this reviewer.
Overall, the article can be accepted for publication at the ‘Energies of MDPI Journal’ in its present format. Therefore, the reviewer recommends the Editor to accept the revised manuscript for publication.
Reviewer 3 Report
I have reviewed the changes made by the authors. The Manuscript is acceptable. Best wishesAuthor Response
Please see the attachment. Thank you.
